# Effects of Intradialytic Exercise on Dialytic Parameters, Health-Related Quality of Life, and Depression Status in Hemodialysis Patients: A Randomized Controlled Trial

**DOI:** 10.3390/ijerph18179205

**Published:** 2021-08-31

**Authors:** Chia-Huei Lin, Yu-Juei Hsu, Pi-Hsiu Hsu, Yi-Ling Lee, Chueh-Ho Lin, Meei-Shyuan Lee, Shang-Lin Chiang

**Affiliations:** 1Nursing Department, Tri-Service General Hospital, Taipei 11490, Taiwan; andyy520@mail.ndmctsgh.edu.tw; 2School of Nursing, National Defense Medical Center, Taipei 11490, Taiwan; 3Nephrology Division, Tri-Service General Hospital, Taipei 11490, Taiwan; yujuei@gmail.com; 4School of Medicine, National Defense Medical Center, Taipei 11490, Taiwan; 5Dialysis Center, Tri-Service General Hospital, Taipei 11490, Taiwan; hd@mail.ndmctsgh.edu.tw; 6Dialysis Center, Songshan Branch of Tri-Service General Hospital, Taipei 10581, Taiwan; elin680202@gmail.com; 7Master Program in Long-Term Care & School of Gerontology Health Management, College of Nursing, Taipei Medical University, Taipei 110301, Taiwan; chueh.ho@tmu.edu.tw; 8Center for Nursing and Healthcare Research in Clinical Practice Application, Wan Fang Hospital, Taipei Medical University, Taipei 110301, Taiwan; 9School of Public Health & Graduated Institute of Medical Science, National Defense Medical Center, Taipei 11490, Taiwan; meei.shyuan@msa.hinet.net; 10Department of Physical Medicine and Rehabilitation, Tri-Service General Hospital, Taipei 11490, Taiwan

**Keywords:** depression status, exercise, hemodialysis, intradialytic exercise, health-related quality of life

## Abstract

Exercise is fundamentally important in managing chronic diseases and improving health-related quality of life (HRQL). However, whether intradialytic exercise is safe through assessment of changes in dialytic parameters and has a positive impact on HRQL and depression status of hemodialysis patients requires further research with diverse racial and cultural populations to identify. This study aimed to evaluate the effects of intradialytic exercise on dialytic parameters, HRQL, and depression status in hemodialysis patients. A randomized controlled trial was conducted at a medical center in Northern Taiwan. Sixty-four hemodialysis patients were recruited using stratified random sampling. Participants were randomized into an experimental group (EG, *n* = 32) or a control group (CG, *n* = 32). The EG received a 12-week intradialytic exercise program while the CG maintained their usual lifestyles. Dialytic parameters, HRQL, and depression status were collected at baseline and at 12 weeks. The results indicated no differences in the dialytic parameters from the baseline between both groups. However, the EG had increased HRQL (ß = 22.6, *p* < 0.001) and reduced depression status (ß = −7.5, *p* = 0.02) at 12 weeks compared to the CG. Therefore, a 12-week intradialytic exercise regime is safe and effective in improving HRQL and reducing depression status for hemodialysis patients.

## 1. Introduction

End-stage renal disease (ESRD), the last stage of chronic kidney disease due to its irreversible loss of renal function, is an emerging global public health problem [1,2]. Nearly 90% of ESRD patients require regular hemodialysis as renal replacement therapy for maintaining survives [1]. However, ESRD patients on hemodialysis maintenance with associated multi-comorbidities (i.e., cardiovascular disease, renal bone disease, and anemia), experience an impaired health-related quality of life (HRQL) [2,3], and increased depression status due to the high burden of somatic symptoms and psychological distress [2,4,5]. In addition, both HRQL and depression status has been reported to be associated with mortality in hemodialysis patients after adjusting for age, gender, race, the primary cause of kidney failure, dialysis vintage, and baseline depression history [4,5]. Therefore, developing strategies to improve HRQL and reduce depression status in hemodialysis patients is an imperative issue.

Physical inactivity, a prominent contributor to the deterioration of physical function among hemodialysis patients [6], has been linked to adverse clinical outcomes such as high morbidity rate and mortality [3,7]. Impairments from complications including fluid retention, anemia, and accumulated uremic toxins are attributable to the disease of ESRD itself, hemodialysis (e.g., the long periods of immobility during treatments and post-dialysis fatigue), and particularly, physical inactivity [6,7]. A vicious circle hence arises between physical inactivity and morbidity in hemodialysis patients.

A systematic review advocated that reduction in physical function of hemodialysis patients can be partially reversed with regular exercise over at least eight weeks [8,9]. In addition, engagement in regular exercise is associated with a decreased mortality risk [3,10], improved HRQL, and reduced depression status among hemodialysis patients [8,11]. Therefore, the National Kidney Foundation and clinical practice guidelines recommend that hemodialysis patients should increase their physical activity levels and make regular exercise a part of the strategic management of hemodialysis treatment [12]. Unfortunately, most hemodialysis patients exhibit low physical activity levels [13,14] for various reasons. Numerous known barriers, involving fear of injuries, discomfort, safety concerns, limited leisure time, symptoms of debilitation, and intolerance of exercise, make most hemodialysis patients experience difficultly in participating in regular exercise [15,16,17]. The most commonly reported barriers are fatigue on dialysis days and shortness of breath on non-dialysis days [15]. In addition, fatigue and muscle weakness after a long period of hemodialysis results in deconditioning, which might further reduce exercise tolerance, compliance/adherence to exercise, and overall motivation [18].

Despite regularly implemented supervised exercise interventions may not be entirely feasible for most chronic diseases in clinical practice, it has been reported to be more effective than home-based or community-based exercise training [19,20]. Hemodialysis patients must receive dialysis treatments regularly two to three times per week at a hospital or dialysis facility to maintain their lives. Therefore, adding exercise to a regular hemodialysis visit would be feasible and an optimal choice for hemodialysis patients under the assumption that it would increase exercise adherence. However, hemodialysis patients experience high cardiovascular and all-cause mortality, particularly during hemodialysis treatments. According to reports from the US Renal Data System database, two-thirds of cardiac deaths are attributed to an arrhythmia-related sudden death, making up 26% of mortality among hemodialysis patients [21]. Although many triggers of sudden death have been identified, such as acute myocardial ischemia, autonomic imbalance, increased sympathetic activity, history of hypertension, and diabetes, the higher risk of sudden death among hemodialysis patients appears to accelerate after dialysis initiation [21,22]. Therefore, whether a moderate-intensity intradialytic exercise intervention is safe requires further investigation, particularly through evidence with diverse racial and cultural groups. The clinical signs and characteristics (i.e., electrolyte imbalances, anemia, and hyperparathyroidism) frequently presented in hemodialysis patients could explain partial mechanisms of arrhythmia, as suggested by the relationship between sudden cardiac death and hemodialysis [23]. In addition, a decreased glomerular filtration rate (GFR) has been proposed to cause endocardial as well as diffuse myocardial fibrosis that could enhance the risk of life-threatening ventricular arrhythmias [24]. Hence, our study aimed to confirm the safety of a 12-week intradialytic exercise program by investigating the changes of serum chemistries, serum electrolytes, and GFR from baseline.

So far, whether intradialytic exercise is safe requires more research with diverse racial and cultural populations to identify, since previous reports evaluated the adverse or accident events during exercise training to determine the safety and particularly limited evidence were found in the Asian population [25]. In addition, inconsistent effects of intradialytic exercise on HRQL are found [26,27,28] and little is known about the effects of a 12-week intradialytic exercise regime on cardiometabolic factors combined with HRQL and depression status. Therefore, the current study aimed to determine the effects of a 12-week intradialytic exercise program on dialytic parameters, cardiometabolic factors, HRQL, and depression status in hemodialysis patients.

## 2. Materials and Methods

### 2.1. Design

A randomized controlled trial with a two parallel-group design was conducted. Eligible ESRD patients undergoing maintenance hemodialysis for at least six months were randomized into either an experimental group (EG) or a control group (CG). The EG received a 12-week intradialytic cycling exercise and the CG maintained their usual lifestyles. Outcome measures including dialytic parameters (i.e., serum chemistries, serum electrolytes, intact-parathyroid hormone [IPTH], and estimated glomerular filtration rate [eGFR]), cardiometabolic factors (i.e., resting heart rate, systolic/diastolic blood pressure, fasting blood glucose, cholesterol, and triglycerides), and Uric acid), HRQL, and depression status were collected at pre- (baseline) and post- (12 weeks) intervention.

### 2.2. Participants

Potential hemodialysis patients were recruited from the hemodialysis center at a medical center in Northern Taiwan between June 2019 and December 2019. One hundred sixty-three patients receiving regular hemodialysis (97 patients treated on odd weekdays [Monday, Wednesday, Friday] and 66 patients treated on even weekdays [Tuesday, Thursday, Saturday]) were initially approached. Those who agreed to participate were then screened for eligibility by a nephrologist. Inclusion criteria were: (1) ESRD patients on hemodialysis maintenance; (2) aged 20 to 80 years; (3) able to speak and understand Mandarin; (4) had received regular treatment with hemodialysis (3 times/week) for at least six months, and (5) agreed to be randomized into one of the two groups. Exclusion criteria included lower limb disabilities, hospitalized patients, treatment with peritoneal dialysis, received hemodialysis less than three times/week, a history of recent acute myocardial infarction, unstable angina, uncontrolled arrhythmia, acute stroke, a hospitalization experience within the past six months, diagnosed cancer, and a mental illness, especially depression.

G*Power (Germany, version 3.1.9) software was applied for sample size estimation [29]. Based on an analysis of variance (ANOVA)-repeat measures (within-between interactions), a statistical power of 0.8, an effect size of 0.25, a significant level of 0.05, and the number of measurements at 2, we calculated that 26 participants in each group would be required [30]. By taking into account a possible attrition rate of 15–20%, the target sample size was set at 30–32 per group.

### 2.3. Study Cohorts and Interventions

Considering that having two groups in the same area for dialysis treatment combined with intradialytic exercise may result in bias, we used stratified random sampling to place participants into either the EG and CG and separated them into different treatment schedules. Therefore, 64 patients were randomly selected from 112 eligible patients (60 were treated on odd weekdays and 52 were treated on even weekdays) and allocated them with a 1:1 randomization ratio into the EG (*n* = 32, treated on odd weekdays) and the CG (*n* = 32, treated on even weekdays) by the research project investigator. Randomization was performed using sealed opaque envelopes which were opened by a research nurse.

The exercise protocol for the EG was prescribed by a rehabilitation physician and followed the principle based on the American College of Sports Medicine guidelines including frequency, intensity, time, and type [31]. (1) Frequency: received intradialytic, lower-limb, cycling exercise three times per week on alternative days (Monday, Wednesday, Friday) for 12 weeks (36 times) at the hemodialysis center of the medical center and supervised by hemodialysis nurses and a research nurse who had more than 10 years of exercise training experience. (2) Intensity: the intensity of exercise was set at 12–14 (moderate-intensity: somewhat hard or reports of feeling a little bit tired but still ok to continue) based on the Borg’s Perceived Exertion Rating Scale (a rating of 6 perceiving “no exertion at all” to 20 perceiving a “maximal exertion” of effort) [32]. Appropriate speed and grade of resistance were adjusted to achieve the required intensity. (3) Time: the duration of each exercise session consisted of a 5-min warm-up, 20-min endurance, and 5-min cool-down phase. The exercise occurred at least one to two hours after a meal. (4) Type: lower-limb ergometer (WP-698, Magnetic Mini Bike, Taiwan) was used for intradialytic cycling exercise in the supine position. Each session of intradialytic exercise started at 30 min after the beginning of hemodialysis when the hemodynamic stability of patients was confirmed (without complaint of chest pain, dyspnea, pallor, diaphoresis, or dizziness; had systolic pressure >200 mmHg or diastolic pressure >120 mmHg; had a decrease in systolic pressure of >10 mmHg compared to the systolic pressure at rest, or requested stopping the exercise). Participants in the CG maintained their usual lifestyles and regular hemodialysis.

### 2.4. Measures

Eligible patients who agreed to participate were invited to the local medical center for pre- (baseline) assessment. Data were collected using structured interviews with questionnaires (sociodemographics, lifestyle factors, HRQL, and depression status), blood analyses, and blood pressure measures at baseline and 12 weeks by a separate research nurse, blinded to the group assignment.

### 2.5. Dialytic Parameters

The trial assessed the safety of a 12-week intradialytic exercise program as compared with conventional treatment in hemodialysis patients through the stability or changes of dialytic parameters at pre- and post-intervention. We evaluated whether dialytic parameters including serum chemistries (red blood cell [count/uL], hemoglobin [g/dL], hematocrit [%], mean corpuscular volume [fL], albumin [g/dL], GPT [IU/L], GOT [IU/L], blood urea nitrogen [BUN, mg/dL], creatinine [Cr, mg/dL]), serum electrolytes (sodium [Na, mEq/L], potassium [K, mEq/L], calcium [Ca, mg/dL], phosphate [P, mg/dL]), IPTH [pg/mL]), and eGFR, [mL/min1.73m^2^]) in the EG were different from the CG. All of the dialytic parameters were analyzed at the clinical laboratory of the local medical center, where was certified by the College of American Pathologists. The eGFR, calculated by the equation: 186 × (Creatinine/88.4) − 1.154 × (age) − 0.203 × (0.742 if female) × (1.210 if black), has been recognized as an indicator for facilitating the detection, evaluation, and management of chronic kidney disease [33].

### 2.6. Cardiometabolic Factors

The positive impact of regular moderate- to vigorous-intensity aerobic exercise on cardiometabolic health has been well-documented [34,35]. Therefore, the cardiometabolic factors, consisting of resting heart rate (beat/min), blood pressure (systolic and diastolic blood pressure [mmHg]), fasting blood glucose (mg/dL), serum lipids (cholesterol [mg/dL] and triglyceride [mg/dL]), and uric acid (mg/dL) were assessed at baseline and 12 weeks, as secondary outcomes. Resting heart rate and blood pressure were obtained after participants had been seated quietly for three to five minutes, using an electronic blood pressure monitor device (Terumo, ESP2000, Tokyo, Japan).

### 2.7. Health-Related Quality of Life

The well-valid and reliable Medical Outcomes Study Short-Form 36 (SF-36), consisting of 36 items and eight subscales (bodily pain [2 items], general health [5 items], mental health [5 items], physical function [10 items], role function limitation due to emotional problems [role-emotional, 3 items], role function limitation due to physical conditions [role-physical, 4 items], social function [2 items], and vitality [4 items]) except one item for health transition was used to assess HRQL [36]. In addition to the eight subscales, the total mean score of HRQL was measured to evaluate the overall HRQL. Higher scores ranged from 0 to 100 presented better HRQL. Cronbach’s alpha of the total scale in the present study was 0 93.

### 2.8. Depression Status

The 21-item, self-rated Beck Depression Inventory (BDI) [37] with good reported validity and reliability [38] was applied to measure participants’ depression status. Scores ranged from 0–63 and higher scores indicated higher depression status. The BDI, comprising of emotional (5 items), cognitive (7 items), and somatic (9 items) categories, can also be used to screen depressive symptoms as minimal depression (0–9), mild depression (10–18), moderate depression (19–29), and severe depression (30–63). Cronbach’s alpha of the scale in the present study was 0.92.

### 2.9. Ethical Consideration

Institutional review board approval (TSGHIRB: 1-108-05-070) was obtained from Tri-Service General Hospital in Taipei, Taiwan. This trial has been registered on the “ClinicalTrials.gov” (NCT04990154). All participants were invited to join the study after giving informed consent and were assured that their participation was entirely voluntary and that they could withdraw at any time.

### 2.10. Data Analysis

Statistical analyses were performed by SPSS version 16.0 (SPSS Inc., Chicago, IL, USA). Descriptive statistics including means, standard deviation (SD), and percentages (%) were used to display the study participants’ sociodemographics, clinical information, and lifestyle characteristics. Student’s t-tests and chi-square tests were used to compare the pre- and post-intervention differences between groups. Paired *t*-tests were applied to compare differences between pre- and post-tests. Generalized estimating equations (GEEs) for longitudinal data/repeat measures were applied to appraise the intervention effects of the two groups by significant interactions of group and time (group × time) as it can be used to evaluate intervention effects under adjustment for potential confounding factors [39]. Both the quantile-quantile plot and Shapiro–Wilk test were used to determine the normality of outcome variables studied. In addition, G*Power (Germany, version 3.1.9) software were also applied for the calculation of the post-hoc effect size under the sample size of 64 and resulted in an effect size of 0.35 [25]. An intent-to-treat analysis was applied to provide unbiased comparisons among the treatment groups and avoid the effects of patient dropouts. The last-observation-carried -forward method of data imputation was adopted to handle missing data. All of the statistical analyses were two-tailed and *p* < 0.05 was considered statistically significant.

## 3. Results

### 3.1. Baseline Characteristics of Participants

One hundred sixty-three patients were initially approached. Of these, 15 participants declined to participate (due to anticipated discomfort and fatigue from intradialytic exercise) and 36 were excluded. Of the remaining 112 participants, 64 were randomly selected and assigned: 32 (50%) to the EG (treated on odd weekdays) and 32 (50%) to the CG (treated on even weekdays). Of the 64 randomized participants, 57 (89%) completed all data collection (29 in the EG and 28 in the CG). The reasons for not completing the study were withdrawal from the study due to fatigue during intradialytic exercise (*n* = 1), suffering from knee osteoarthritis with severe pain (*n* = 1), and loss to follow-up due to hospitalization for coronary artery disease (*n* = 4) and pneumonia (*n* = 1) (Figure 1). The last-observation-carried-forward method of data imputation was used for intent-to-treat analysis. Hence, sixty-four participants were included in the data analysis.

Table 1 shows the sociodemographic characteristics, comorbidities, and lifestyle factors. The two groups did not differ in sociodemographics, comorbidities, and lifestyle factors. The baseline dialytic parameters, cardiometabolic factors, HRQL, and depression status in the two groups are shown in Table 2.

### 3.2. Outcome Evaluation

#### 3.2.1. Dialytic Parameters

The descriptive and univariate analyses of the outcome evaluation are shown in Table 2. There were no differences between the two groups in all of the baseline dialytic parameters. Given that participants in the EG had higher (*t* = 2.43, *p* = 0.02) albumin levels than those in the CG at 12 weeks, both EG and CG had no significant change in all of the dialytic parameters after 12 weeks. When the group × time interaction was examined based on the GEE analyses (Table 3), all of the dialytic parameters in the EG had no changes as compared to the CG.

#### 3.2.2. Cardiometabolic Factor

There were no differences between the two groups in all of the baseline cardiometabolic factors (Table 2). Given that the EG had reduced systolic blood pressure (*t* = −3.03, *p* = 0.004) at 12 weeks as compared to the CG, both the EG and CG had no significant change in all of the cardiometabolic factors after 12 weeks. When the group × time interaction was examined based on the GEE analyses (Table 3), all of the cardiometabolic factors in the EG had no changes as compared to the CG.

#### 3.2.3. Health-Related Quality of Life

There were no differences between the two groups in the baseline HRQL including the total mean score of HRQL and the eight subscales (Table 2). After the 12-week intradialytic exercise, the EG had an increased total mean score of HRQL, bodily pain, general health, mental health, physical function, role-physical, social functioning, and vitality. The CG had no changes within these from baseline. In addition, participants in the EG had a higher total mean score of HRQL and with most subscales except the bodily pain compared to the CG. When the group × time interaction was examined based on the GEE analyses, participants in the EG had a greater increase in total mean score of HRQL (ß = 22.6, *p* < 0.001), general health (ß = 19.2, *p* = 0.004), mental health (ß = 17.7, *p* = 0.001), physical function (ß = 14.5, *p* = 0.02), role-emotional (ß = 28.9, *p* = 0.04), and role-physical (ß = 63.7, *p* < 0.001) at 12 weeks as compared to those in the CG after adjusting for sociodemographic characteristics, comorbidities, and lifestyle factors (Table 3).

#### 3.2.4. Depression Status

There were no differences between the two groups in baseline depression status (Table 2). After the 12-week intradialytic exercise, the EG had a significantly lower depression status while the CG rendered no changes. The significant group × time interaction for depression status revealed that the EG had a greater decrease in depression status at 12 weeks as compared to the CG (ß = −7.5, *p* = 0.02) after adjusting for sociodemographic characteristics, comorbidities, and lifestyle factors (Table 3).

## 4. Discussion

The results of the current study demonstrated that a 12-week intradialytic exercise intervention is effective in improving HRQL and decreasing depression status among hemodialysis patients, but presents no differences in the dialytic parameters, indicating the intradialytic exercise regime is safe for hemodialysis patients. Given that a 12-week, moderate-intensity intradialytic exercise program had no additional benefit in cardiometabolic factors, our study results add to the literature illustrating that intradialytic exercise has a positive impact on HRQL and depression status without remarkable adverse events in hemodialysis patients, highlighting its clinical benefit when it is provided in combination with hemodialysis treatments in ESRD patients.

During hemodialysis, large fluid volumes are extracted followed by delayed reuptake of water from the interstitial space which leads to an inability to normalize arterial plasma volume. This causes a decline in cardiac output and reduces myocardial and systemic perfusion in 20–30% of ESRD patients [40]. A recent review advocates the need for more research to assess the safety of intradialytic exercise for hemodialysis patients among diverse cultures or regions since most studies have been conducted with Western populations [25,26]. In addition, previous studies assessed the adverse or accident events during exercise training to determine the safety [25,26] instead of dialytic parameters such as serum chemistries, electrolytes, and GFR. The current study thus examined the effects of intradialytic exercise on the dialytic parameters, including serum chemistries (red blood cell, hemoglobin, hematocrit, mean corpuscular volume, albumin, GPT, GOT, BUN, Cr), serum electrolytes (sodium, potassium, calcium, phosphate), and IPTH in Asian population to determine its safety. Our findings showed that after a 12-week intradialytic exercise regime, the EG had no changes in all of the dialytic parameters as compared to the CG. This finding is consistent with previous research that has shown how intradialytic exercise had no significant effect on serum phosphate levels and PTH [25] as well as the serum calcium and hemoglobin levels [41]. In addition, some dialytic parameters such as hemoglobin and electrolytes might be influenced by an individual’s nutritional status and dietary patterns. Future studies, therefore, must investigate these effects under standardized diet formulas in both groups.

A recent study also revealed how exercise benefits non-dialysis patients with chronic kidney disease by increasing eGFR [42]. However, there are scant references regarding its effects in hemodialysis patients or patients who participated in intradialytic exercise. In turn, this study evaluated the effects of intradialytic exercise on glomerular filtration rate, revealing that the EG had no changes in eGFR compared to the CG. The renal functions of the participated hemodialysis patients who reported an average hemodialysis duration of 6.2–6.7 years in this study indicate an irreversible progression, contributing to such an unchangeable result of the glomerular filtration rate. However, during exercise, the distribution of cardiac output shifting to the skeletal muscles would cause a decrease in renal blood perfusion. Additionally, the higher intensity of the exercise, the lower the proportionate distribution of cardiac output is found [42]. Whether receiving hemodialysis combined with intradialytic exercise may aggravate the reduction of renal perfusion during the training process remains for further research to identify. Our current study evaluated the effects of intradialytic exercise on the changes of glomerular filtration rate among hemodialysis patients, confirming the safety of a 12-week, moderate intradialytic exercise for hemodialysis patients. These findings are in line with previous reports [27,43]. In addition, during the 12-week intradialytic exercise, no adverse events, including intradialytic hypotension, were observed in our study except for two patients reporting exercise-related limb pain, which was consistent with previous reports [26,27]. We conjecture that acute physiological responses to intradialytic exercise may help increase blood volume by inducing greater reuptake of blood from tissue, contributing to hemodynamic stability [44]. Sheng and his colleagues identified that intradialytic exercise can even improve *Kt/V*, proving the safety of the exercise regimen [43]. Therefore, the current study used different parameters to add literature confirming the safety of a 12-week moderate-intensity intradialytic exercise program in hemodialysis patients with a preliminary result investigating the association between intradialytic exercise and eGFR, providing considerations for future research to design alternative or various intradialytic exercise prescriptions in hemodialysis patients.

Substantial evidence concluded that exercise significantly improved patients’ cardiometabolic health [45,46]. However, the effects of intradialytic exercise on cardiometabolic health remain limited, particularly involving Asian populations. Hence, we examined the effects of a 12-week aerobic intradialytic exercise on cardiometabolic health of Taiwanese hemodialysis patients and revealed no additional benefits in cardiometabolic factors including systolic/diastolic blood pressure, resting heart rate, fasting blood glucose, or serum lipids (cholesterol, triglyceride, and uric acid), which are in line with a previous study [43]. The reasons explaining these insignificant effects can be complicated. A prolonged exercise program over at least six months tends to have a positive impact on cardiometabolic health for hemodialysis patients [43]. In addition, a moderate-to-vigorous exercise regime would be a major predictor to decreased insulin resistance and improved cardiometabolic status [47]. Given that the intensity of exercise was moderate in our study, a subjective rating of intensity through individuals’ perceived exertion is not entirely objective and possibly led to overestimations of the intensity of the exercise. Future studies are thus recommended to apply objective instruments to accurately measure the intensity of exercise, providing exercise training with adequate intensity.

Several previous studies have concluded that combining aerobic and resistance exercise tends to have a positive impact on cardiometabolic factors since combined aerobic exercise and strength training reveals more favorable results regarding improved cardiorespiratory fitness [48]. Cardiorespiratory fitness is a component of physiological fitness that relates to the circulatory and respiratory system′s ability to supply oxygen during sustained physical activity. However, several causes such as anemia, muscular atrophy, hypervolemia, cardiac dysfunction, and physical deconditioning lead hemodialysis patients to an extremely low level of cardiorespiratory fitness [48]. Participants in this study presented a relatively aging population, thus this problem may have been exaggerated.

Hemodialysis patients experience a heavy burden of symptoms and are more inactive, leading to poor functional capacity and a decreased HRQL [43,49], which our study confirms, particularly regarding a lower score of general health, role function limitation due to physical conditions (role-physical), and vitality. While a recent systematic review advocated that regular exercise may reduce depression and fatigue in hemodialysis patients [26], more randomized controlled trials that focus on different exercise regimens are required. Gomes and his research team examined the effects of different intradialytic exercise training modalities among hemodialysis patients and revealed that aerobic exercise alone was not significantly associated with physical function (i.e., aerobic capacity) and HRQL [28]. However, other research teams had different conclusions [25,27]. Given that substantial evidence examined the effects of intradialytic exercise on HRQL, inconsistent results were found, particularly result from different exercise modalities. To understand the effects of intradialytic exercise on HRQL in a specific exercise prescription is required and better to compare, identifying an optima exercise prescription for hemodialysis patients. Hence, our study examined the effects of a 12-week, moderate-intensity intradialytic exercise on HRQL and depression status in hemodialysis patients to provide further evidence for fulfilling the knowledge gap. Accordingly, we found an effectively positive impact on HRQL and depression status, involving the overall mean score, general health, mental health, physical function, and role function due to emotional problems (role-emotional), and role function due to physical conditions (role-physical), except for the subscales of bodily pain, social functioning, and vitality. In a previous systematic review, which is inconsistent with our study results, only the physical aspects of HRQL were improved rather than the mental aspects of HRQL after receiving intradialytic exercise [26]. Another review also suggested that aerobic exercise alone was not associated with HRQL improvement [28]. These differing results might attribute to a different follow-up period, exercise prescription (frequency, intensity, type, and time), or implementation of the exercise. In addition, possible reasons for the changes in the mental domains of HRQL or psychological health (decreased depression) may be associated with factors contributing to patients’ overall mental and emotional states such as disability, degree of dependence [50], the burden of the disease itself, financial problems resulting from unemployment, or available family/social support. Further research is therefore recommended to adjust for these potential confounding factors to accurately identify the mediators of HRQL and intradialytic exercise among hemodialysis patients.

Depression, which is associated with morbidity and mortality in hemodialysis patients [4,5,51], has been identified as the most prevalent psychological problem in hemodialysis patients. In the current study, we found that a 12-week intradialytic exercise is effective in decreasing depression status among hemodialysis patients. Whether this finding is attributed to frequent social interactions with medical staff during hemodialysis treatments, increased vitality, or increased confidence in the management of the disease is not known. Further efforts are required to illustrate this mechanism.

Intradialytic exercise for hemodialysis patients has emerged in recent studies. However, evidence remains insufficient and requires more high-quality clinical trials with diverse racial and cultural groups to clarify and reach conclusions. Therefore, we provided results pertaining to an Asian population to support this research gap. However, several limitations must be acknowledged in this study, including (1) a lack of long-term follow-up evaluation, (2) limited generalizability due to the sampling method (the EG and CG were only randomly selected from the treatment groups of specific days in the week) given that all of the sociodemographic factors, comorbidities, and lifestyle factors were similar between the groups, (3) a limited geographic region where the study was conducted, and (4) the use of a single urban medical center which limits its generalizability for rural areas. Hence, these findings must be interpreted with caution, and larger sample sizes, as well as more repeated evaluations with a longer follow-up period, are required. The strengths of this study include its random allocation design and the high rate of completion (89%) by the participants who were diagnosed with ESRD with a mean hemodialysis treatment period of 6.5 years. Therefore, since intradialytic exercise rendered better compliance/adherence among hemodialysis patients compared to protocols implemented outside hemodialysis centers [48], we recommend that intradialytic exercise combined with hemodialysis treatment should be integrated into clinical settings for hemodialysis patients.

## 5. Conclusions

A 12-week aerobic intradialytic exercise regime is safe and feasible for hemodialysis patients. Adding intradialytic exercise into the hemodialysis process has positive effects on improved health-related quality of life and decreased depression in hemodialysis patients. Further study designs are suggested to evaluate whether a longer duration, higher intensity, or different mode of exercise (such as a combination of aerobic exercise and strength/resistance training) benefits the dialytic parameters and cardiometabolic factors in hemodialysis.

## Figures and Tables

**Figure 1 ijerph-18-09205-f001:**
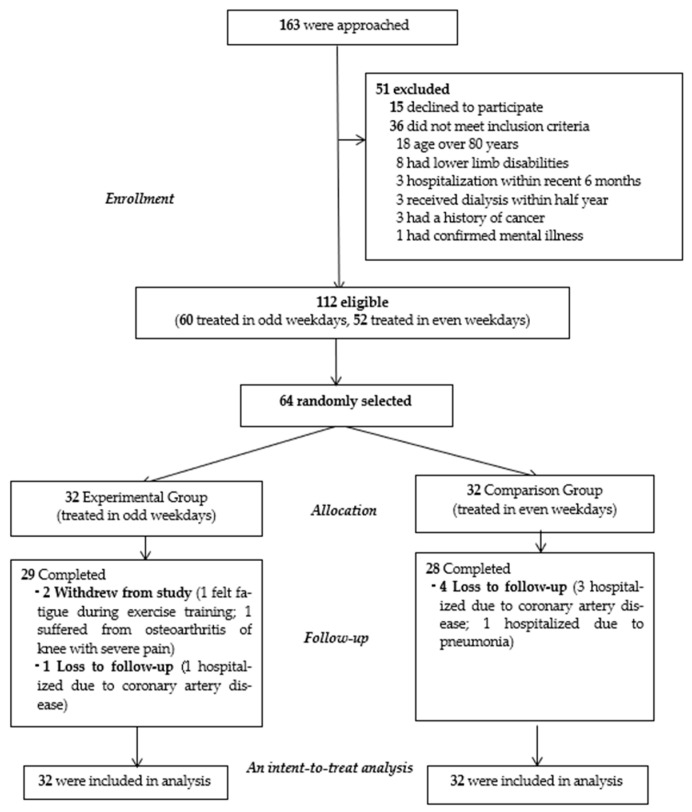
CONSORT Diagram of Participants’ Flow Through the Trial.

**Table 1 ijerph-18-09205-t001:** Comparisons of demographics and comorbidity characteristics between groups.

Variables	EG	CG	*t/x* ^2^	*p*
	*n* = 32	*n* = 32		
**Sociodemographic Characteristics**				
Age (year), *mean (SD)*	62.0 (9.5)	62.1 (12.3)	0.03	0.97
Gender (male), *n (%)*	22 (68.8)	19 (59.4)	0.61	0.60
Marital status (married), *n (%)*	28 (87.5)	23 (71.9)	2.41	0.12
Education (more than high school), *n (%)*	21 (65.6)	25 (78.1)	1.24	0.27
Currently employed, *n (%)*	10 (31.2)	7 (21.9)	0.72	0.40
Body mass index (Kg/m^2^), *mean (SD)*	23.4 (3.7)	23.4 (4.5)	−0.06	0.95
Duration of hemodialysis (year), *mean (SD)*	6.7 (5.7)	6.2 (5.1)	0.39	0.70
**Comorbidities**				
Hypertension, *n (%)*	28 (87.5)	24 (75.0)	1.64	0.34
Type 2 diabetes, *n (%)*	14 (43.8)	19 (59.4)	1.56	0.21
Hyperlipidemia, *n (%)*	5 (15.6)	6 (18.8)	0.11	0.74
Cardiovascular disease, *n (%)*	4 (12.5)	11 (34.4)	4.27	0.08
Metabolic syndrome, *n (%)*	6 (18.8)	7 (21.9)	0.10	0.76
**Lifestyle Factors**				
Smoking, *n (%)*	3 (9.4)	0 (0)	3.15	0.24
Drinking, *n (%)*	0 (0)	1 (3.1)	1.02	1.00

Note: *n*, number; SD, standard deviation; EG, experimental group; CG, comparison group.

**Table 2 ijerph-18-09205-t002:** Differences of outcome indicators between groups at baseline and after the intervention.

Variables	EG (*n* = 32)			CG (*n* = 32)			Baseline		12-Week	
	Baseline	12-Week	*p*	Baseline	12-Week	*p*	EG vs. CG		EG vs. CG	
							*t*	*p*	*t*	*p*
**Dialytic parameters**										
*Serum chemistries*										
Red blood cell (×10^6/uL)	3.5 (0.6)	3.4 (0.6)	0.58	3.6 (0.7)	3.6 (0.9)	1.00	−0.46	0.647	−0.83	0.41
Hemoglobin (g/dL)	10.1 (1.4)	9.9 (1.1)	0.51	10.3 (1.3)	10.4 (1.3)	0.90	−0.676	0.50	−2.17	0.11
Hematocrit (%)	30.5 (4.2)	29.6 (3.6)	0.37	31.4 (4.4)	31.6 (5.6)	0.82	−0.827	0.411	−1.74	0.09
Mean corpuscular volume (fL)	87.9 (9.4)	88.1 (9.6)	0.95	89.0 (8.6)	93.1 (17.9)	0.31	−0.45	0.654	−1.35	0.18
Albumin (g/dL)	3.9 (0.3)	4.0 (0.3)	0.65	3.9 (0.3)	3.8 (0.2)	0.20	0.65	0.52	2.43	0.02
GPT (IU/L)	13.2 (4.2)	14.2 (4.5)	0.37	14.7 (7.1)	14.5 (7.4)	0.89	−1.05	0.30	−0.18	0.86
GOT (IU/L)	13.3 (4.9)	15.0 (6.8)	0.25	14.2 (6.1)	15.4 (6.6)	0.45	−0.68	0.50	−0.26	0.80
BUN (mg/dL)	65.6 (18.1)	70.4 (20.6)	0.33	65.8 (16.5)	70.5 (25.6)	0.39	−0.04	0.97	−0.02	0.99
Cr (mg/dL)	10.6 (2.2)	10.4 (2.3)	0.72	9.7 (2.0)	10.1 (2.1)	0.43	1.69	0.10	0.49	0.63
*Serum electrolytes*										
Na (mEq/L)	137.6 (3.0)	137.5 (2.4)	0.92	137.4 (2.8)	142.0 (24.5)	0.29	0.31	0.76	−1.04	0.30
K (mEq/L)	4.8 (0.8)	4.6 (0.6)	0.29	4.6 (0.7)	4.7 (1.1)	0.66	1.11	0.27	−0.40	0.69
Ca (mg/dL)	9.2 (1.2)	9.3 (1.1)	0.91	9.3 (1.0)	9.0 (0.9)	0.23	−0.13	0.90	1.16	0.25
P (mg/dL)	5.2 (1.4)	4.9 (1.2)	0.48	4.8 (1.2)	5.0 (1.3)	0.69	1.05	0.30	−0.06	0.95
*IPTH* (pg/mL)	416.3 (351.6)	438.7 (293.9)	0.79	330.9 (335.0)	321.8 (289.2)	0.93	0.81	0.42	1.47	0.15
*eGFR* (mL/min1.73m^2^)	5.0 (1.1)	5.1 (1.2)	0.67	5.4 (1.5)	5.1 (1.3)	0.38	−1.415	0.162	−0.10	0.92
**Cardiometabolic factors**										
Systolic blood pressure (mmHg)	141.5 (20.8)	136.5 (14.4)	0.27	145.1 (8.0)	152.2 (25.5)	0.30	−0.58	0.562	−3.03	0.004
Diastolic blood pressure (mmHg)	75.6 (12.2)	71.7 (7.4)	0.13	73.1 (13.8)	74.8 (14.4)	0.62	0.76	0.451	−1.09	0.28
Resting heart rate (beat/min)	71.6 (8.3)	71.4 (8.5)	0.93	71.4 (9.8)	70.8 (9.7)	0.79	0.08	0.935	0.29	0.77
Fasting blood glucose (mg/dL)	105.4 (29.1)	103.6 (28.3)	0.81	125.8 (86.5)	116.8 (47.2)	0.61	−1.26	0.21	−1.36	0.18
Cholesterol (mg/dL)	156.6 (32.2)	153.8 (31.3)	0.73	156.5 (29.9)	157.4 (32.3)	0.92	0.01	0.99	−0.42	0.68
Triglyceride (mg/dL)	124.6 (127.6)	125.8 (116.2)	0.97	112.3 (62.9)	119.2 (86.5)	0.75	0.43	0.67	0.24	0.82
Uric acid (mg/dL)	6.4 (1.7)	5.7 (1.3)	0.16	6.2 (1.2)	6.0 (1.4)	0.64	0.60	0.55	−0.53	0.60
**HRQL**										
Total mean score	62.8 (17.5)	81.0 (18.7)	<0.001	64.1 (16.9)	58.1 (16.1)	0.15	0.30	0.77	5.27	<0.001
Bodily pain	74.4 (22.5)	90.8 (18.9)	0.003	79.9 (26.6)	79.9 (26.6)	1.00	0.89	0.38	1.89	0.06
General health	44.1 (17.6)	58.6 (20.6)	0.004	48.1 (18.8)	42.7 (19.8)	0.26	0.89	0.38	3.16	0.002
Mental health	63.8 (16.8)	79.8 (18.2)	0.001	70.6 (15.7)	67.4 (14.4)	0.39	1.69	0.10	3.02	0.004
Physical function	72.5 (21.0)	88.3 (15.7)	0.001	75.0 (19.3)	73.9 (19.1)	0.82	0.50	0.62	3.29	0.002
Role-emotional	78.1 (37.5)	90.6 (29.6)	0.14	63.5 (40.9)	44.8 (42.8)	0.08	−1.49	0.14	4.98	<0.001
Role-physical	46.9 (39.0)	85.9 (31.7)	<0.001	50.0 (43.1)	32.0 (39.3)	0.09	0.30	0.76	6.04	<0.001
Social functioning	71.9 (24.4)	88.7 (22.8)	0.01	72.3 (22.6)	70.7 (22.1)	0.78	0.07	0.95	3.20	0.002
Vitality	50.6 (20.6)	65.6 (21.1)	0.01	53.0 (19.1)	53.4 (18.1)	0.92	0.47	0.64	2.48	0.02
**Depression status**	12.8 (9.3)	5.0 (6.8)	<0.001	11.2 (9.8)	12.5 (9.2)	0.58	0.65	0.52	−3.73	<0.001

Note: EG, experimental group; CG, comparison group; GPT, glutamic pyruvic transaminase; GOT, glutamic oxaloacetic transaminase; BUN, blood urea nitrogen; Cr, creatinine; IPTH, intact parathyroid hormone; *eGFR*, estimated glomerular filtration rate; HRQL, health-related quality of life; data are presented as mean (SD); *p*-values were from the paired *t*-test, Student’s *t*-test, or chi-square test.

**Table 3 ijerph-18-09205-t003:** Evaluation of the intervention on dialytic parameters, health-related quality of life, and depression status based on the GEE Analysis.

Variables	Within Group		Between Group		Interaction Group (EG) × Time				Interaction ^a^ Group (EG) × Time			
	Ref: Baseline		Ref: CG		Reference Group: (CG) × Time				Reference Group: (CG) × Time			
	ß	*p*	ß	*p*	ß	95% C.I.		*p*	ß	95% C.I.		*p-Adjusted*
						Lower	Upper			Lower	Upper	
**Dialytic parameters**												
*Serum chemistries*												
Red blood cell (×10^6/uL)	0.001	1.00	−0.08	0.65	−0.08	−0.58	0.42	0.76	−0.03	−0.47	0.42	0.90
Hemoglobin (g/dL)	0.05	0.89	−0.22	0.51	−0.27	−1.13	0.59	0.54	−0.23	−1.03	0.58	0.58
Hematocrit (%)	0.29	0.82	−0.89	0.40	−1.17	−4.27	1.92	0.46	−0.96	−3.86	1.95	0.52
Mean corpuscular volume (fL)	4.07	0.28	−1.09	0.64	−3.90	−12.6	4.83	0.38	−3.82	−11.6	3.96	0.34
Albumin (g/dL)	−0.08	0.18	0.04	0.51	0.12	−0.07	0.30	0.21	0.11	−0.07	0.29	0.21
GPT (IU/L)	−0.25	0.89	−1.53	0.29	1.25	−2.85	5.35	0.55	1.42	−2.16	5.01	0.44
GOT (IU/L)	1.22	0.44	−0.94	0.49	0.50	−3.70	4.70	0.82	0.69	−3.26	4.63	0.73
BUN (mg/dL)	4.66	0.38	−0.19	0.97	0.09	−13.9	14.1	0.99	0.93	−12.2	14.1	0.89
Cr (mg/dL)	0.41	0.42	0.88	0.09	−0.61	−2.07	0.85	0.41	−0.53	−1.75	0.69	0.40
*Serum electrolytes*												
Na (mEq/L)	4.68	0.28	0.22	0.76	−4.74	−13.3	3.77	0.28	−4.71	−12.8	3.34	0.25
K (mEq/L)	0.10	0.65	0.20	0.26	−0.29	−0.84	0.26	0.31	−0.27	−0.80	0.25	0.31
Ca (mg/dL)	−0.29	0.21	−0.03	0.90	0.33	−0.39	1.04	0.37	0.31	−0.38	1.00	0.38
P (mg/dL)	0.13	0.69	0.34	0.28	−0.36	−1.23	0.52	0.42	−0.30	−1.14	0.54	0.49
*IPTH* (pg/mL)	−9.09	0.93	85.38	0.40	31.49	−220.2	283.2	0.81	3.5	−227.5	234.5	0.98
*eGFR* (mL/min1.73m^2^)	−0.31	0.37	−0.47	0.15	0.44	−0.44	1.32	0.33	0.40	−0.38	1.18	0.32
**Cardiometabolic factors**												
Systolic blood pressure (mmHg)	−7.06	0.28	−15.7	0.002	12.1	−3.44	27.6	0.13	−12.5	−26.5	1.37	0.08
Diastolic blood pressure (mmHg)	1.75	0.61	2.47	0.44	−5.59	−14.0	2.77	0.19	−5.80	−13.2	1.59	0.12
Resting heart rate (beat/min)	−0.66	0.79	0.19	0.93	0.47	−5.74	6.68	0.88	0.54	−5.29	6.37	0.86
Fasting blood glucose (mg/dL)	−8.94	0.60	−20.3	0.20	7.16	−29.2	43.5	0.70	5.80	−27.2	38.8	0.73
Cholesterol (mg/dL)	0.86	0.92	0.11	0.99	−3.67	−26.8	19.4	0.76	−8.69	−29.4	12.0	0.41
Triglyceride (mg/dL)	6.95	0.74	12.3	0.63	−5.72	−78.4	67.0	0.88	−4.18	−70.3	61.9	0.90
Uric acid (mg/d)	−0.22	0.64	0.22	0.25	−0.54	−1.83	0.75	0.41	−0.57	−1.78	0.65	0.36
**HRQL**												
Total mean score	−6.0	0.14	−1.3	0.76	24.2	12.4	36.0	<0.001	22.6	11.2	34.0	<0.001
Bodily pain	0.0	1.00	−5.5	0.37	16.4	0.1	32.7	0.05	12.8	−2.0	27.5	0.09
General health	−5.5	0.25	−4.1	0.36	20.0	6.9	33.1	0.003	19.2	6.3	32.1	0.004
Mental health	−3.3	0.38	−6.9	0.09	19.3	8.1	30.4	0.001	17.7	7.5	27.8	0.001
Physical function	−1.1	0.82	−2.5	0.61	16.9	4.0	29.7	0.01	14.5	2.1	27.0	0.02
Role-emotional	−18.8	0.07	14.6	0.13	31.3	5.3	57.2	0.02	28.9	2.1	55.8	0.04
Role-physical	−18.0	0.08	−3.1	0.76	57.0	30.8	83.3	<0.001	63.7	36.3	91.1	<0.001
Social functioning	−1.6	0.78	−0.4	0.95	18.4	2.7	34.0	0.02	13.9	−1.4	29.1	0.08
Vitality	0.5	0.92	−2.3	0.63	14.5	1.1	28.0	0.04	10.2	−3.4	23.7	0.14
**Depression status**	1.3	0.57	1.6	0.51	−9.1	−15.1	−3.1	0.003	−7.5	−13.8	−1.3	0.02

Note: EG, experimental group: exercise training; CG, control group: received routine usual care; ß: Regression coefficient; Analyses were performed by GEE models, with a Group × Time interaction term characterizing the intervention effect of interest; *p*-adjusted, **^a^** models were adjusted for sociodemographic characteristics (sex, age, educational level, marital status, current employment, body mass index, duration of hemodialysis), comorbidities (hypertension, type 2 disease, heart disease, hyperlipidemia, metabolic syndrome) and lifestyle factors (smoking and drinking).

## Data Availability

The datasets used and/or analyzed during the current study are available from the corresponding author on reasonable request.

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
