# Peer review of "Effects of Intradialytic Exercise on Dialytic Parameters, Health-Related Quality of Life, and Depression Status in Hemodialysis Patients: A Randomized Controlled Trial"

_ijerph, 2021, doi:10.3390/ijerph18179205_

Round 1
Reviewer 1 Report
This is an interesting paper on an important topic for nephrologists and other members of care team. The manuscript presents the findings of a randomized controlled trial (with a two parallel-group) explores the effects of a 12-week intradialytic exercise program on dialytic parameters, cardiometabolic factors, HRQL, and depression status in hemodialysis patients. Introduction includes all necessary information. Material and methods are adequately described. Results are clearly presented, and Discussion is comprehensive. Limitations and conclusion are sufficient. Manuscript is well written. This paper will be a useful addition to the literature.
Congratulations to the authors.
Author Response
Thank you for reviewing our manuscript. Your encouragement is highly appreciated.

Reviewer 2 Report
I would like to congratulate the authors for the work done. I believe that it is a very interesting study and that it can have a great impact on the knowledge about the effects of physical exercise in the population under study.
However, I would like to make some suggestions to the authors, in order to improve the development of the report.
Abstract:
Lines 23-26: “Exercise is fundamentally important in managing chronic diseases and improving health-related quality of life (HRQL) as well as psychological health. However, whether intradialytic exercise is safe and has a positive impact on the HRQL and depression status of hemodialysis patients requires further research.”
Please justify the need for further research.
Introduction:
Lines 47, 53 and 65. I consider it necessary to include citations.
Lines 117-118. From my point of view, it would be necessary to better justify the shortcomings in previous research and what this study aims to contribute.
Materials and Methods:
Lines 155-161. Reflect on whether part of the information should appear in the data analysis section.
Lines 234-250. Add information on number of items per factor, example of factor and internal consistency value for this study.
Lines 251-276. Include the effect size in cases where possible, and data normality analysis.
Please, include information about informed consent and ethical committee approval.
Results:
Table 1. Format. Add top line.
Were follow-up analyzes performed? For example, at 4-8 weeks post exercise?
Discussion:
I suggest starting the discussion by recalling the objective of the study and making a general assessment of the results.
The results are argued and compared with previous studies. However, I suggest reviewing the discussion, since I believe that the justification of the results found could be deepened. That is, increase the information that argues the differences with previous studies or novel aspects of the present study.
Author Response
We are grateful for the opportunity to revise this manuscript. Your valuable comments are well taken and have been very helpful to the clarity of our manuscript.
